# Analysis of Hot Bar Soldering, Insulation Displacement Connections (IDC), and Anisotropic Conductive Adhesives (ACA), for the Automated Production of Smart Textiles

**DOI:** 10.3390/s20010005

**Published:** 2019-12-18

**Authors:** Sebastian Micus, Ivan Kirsten, Michael Haupt, Götz T. Gresser

**Affiliations:** German Institutes of Textile and Fiber Research DITF, 73770 Denkendorf, Germany; ivan.kirsten@auik.de (I.K.); michael.haupt@ditf.de (M.H.); goetz.gresser@ditf.de (G.T.G.)

**Keywords:** smart textiles, joining method, joining technique, joining processes, soldering, bonding, integration of electronics

## Abstract

Despite all the growth forecasts of the smart textiles market, there is no stable automated manufacturing process for attaching classic electronics to textiles. The great amount of manual production steps causes high prices, which slow down market growth. During the production process, the contacting step offers the greatest potential to reduce manual manufacturing steps. For this reason, we have analyzed various contacting methods for electronic parts on conductive yarns that have a high potential for automation. The chosen methods were thermode soldering, insulation–displacement connectors and anisotropic conductive adhesives. In order to ensure reliable mechanical contacting, the samples were tested in a peeling experiment. The examination of the contact resistances took place in the context of a resistance test using four-wire measuring technology.

## 1. Introduction

Current studies confirm that the market volume of smart textiles will grow significantly in the coming years. In addition to the current medical and military areas of application, as well as occupational safety application will spread to the areas of fashion and sport [1]. At the moment, however, we cannot see a real market breakthrough. One of the reasons for the low market penetration of electronic textiles is the high share of manual tasks during their production [2]. Furthermore, the high proportion of manual production steps ensures high prices, which slow down market growth. This is the reason why the German Institute of Textile and Fiber Research (DITF) has searched intensively for automatable contacting methods. In this paper, we have focused on isolated wires. These are inside the textile, meaning it is possible to join components from both sides. Automated methods for contacting printed circuit boards PCBs to isolated wires have not yet been sufficiently investigated in the literature. The production process typically consists of three steps. At first, the wires are uninsulated automatically by a laser. After this, the main contacting process starts, which is investigated in this paper. In order to protect the electronics from environmental influences, the components are encapsulated in an injection molding process [3]. For this reason, we have investigated the mechanical properties of the contacts in a peeling test. This makes it possible to determine whether the connection is robust enough for further production steps.

### 1.1. Hot Bar Soldering

Hot bar soldering is one of several solder processes for contacting PCB to textiles [4]. A pulsed heat thermode is used to join two pre-tinned parts. This technology is suitable for mass production and involves reliable processing conditions and results. It is a very cost-effective process because multiple connections can be made simultaneously and it has a fast temperature ramp-up and cool-down rate [5]. Hot bar soldering was developed for simultaneously soldering many wires (Figure 1a) [6]. It is a re-flow process using a thermode made of titanium [7] or molybdenum. For a defined force application a linear guide with a force gauge is used. By measuring the sinking path, correct positioning of the component and the melting of the solder can be determined [5,6,8,9]. The contacting process of the hot bar soldering system starts by pressing the thermode with a defined force on the contact point. For many processes a force of less than 20 N is sufficient. The thermode heats and cools according to the reflow profile [7]. Overall, thermode soldering is relatively easy to automate. The automated positioning of the contact partners is one of the challenges which can be simplified through positioning pins. Woznicki [10] describes problems with very small pad pitches [10].

### 1.2. Insulation Displacement Connection (IDC)

Insulation displacement connection is a technique based on making an electrical connection by clamping a conductor into a contact [11]. In contrast to crimping, only the conductor is deformed, not the contact. Lehn et al. [12,13] use insulation displacement connectors, especially robbon cable connectors, for the integration of PCDs in textiles. The connections are made within the housing of the connector through the use of a sharp V-shaped contact that cuts through the insulation to connect to the conductor. An assembly with cutters, sockets, pins, PCBs, and electronics provides a big block on the textile without any protection against environment influences. A significant problem with these connectors is the potential of cutting the wire. Lehn himself makes the suggestion to mount IDCs directly on PCBs to obtain a smaller package [12,13]. Martin et al. [14] present a case study of a prototype sensor jumpsuit for motion capture. They integrated isolated wires to the textile. The different sensors were connected to the wires in special areas. The sensors and wires were connected by IDCs [14].

### 1.3. Adhesive Bonding Methods—Anisotropic Conductive Adhesive (ACA)

Linz et al. [15] investigated non-conductive adhesive (NCA) bonding to integrate electronics to textiles. They introduced NCA as a reliable method for contacting rigid electronic modules to embroidered fabric substrates with metal-coated polymer yarns [15]. Furthermore, Linz et al. [16] performed investigations with laser-structured conductive fabrics laminated onto non-conductive fabrics. In the following, they used harsher reliability test conditions and more detailed investigations of non-conductive adhesives to confirm the advancement of this technology [16]. Krshiwoblozki et al. [17] used mechanical reinforced litz wires with and without insulation. They showed the interconnection to woven highly conductive metal-coated polymer yarns. In addition, the selection and investigation of a suitable thermoplastic insulation material was investigated [17]. Hence, the total potential of adhesive bonding methods has not yet been fully realized [18].

## 2. Materials and Methods

In this paper we selected three different contacting methods for which we perceive a high potential for automated integration of electronics on textiles:Hot bar soldering has high potential for the automated production of smart textiles due to the parallel application of pressure and temperature as well as the simultaneous soldering of several places.Insulation displacement connections compensate for poor manufacturing accuracy and the associated positioning problems of textiles with their V shape.Anisotropic conductive adhesive prevents short circuits during the simultaneous curing of nearby contact points and has a much lower thermal impact on electronic parts [19].

In order to be able to compare the possible methods, 30 samples of each contacting method were prepared. The process was carried out on one-layered test boards which were optimized for contacting. The test boards were attached to a knitted tape of polyester fibers (synthetic fibers) with a width of 24 mm and a thickness of 1 mm. Four comparatively large contact points were used for contacting. A gold-plated male connector was manually soldered (Sn96, 5Ag3Cu0, 5 and 3, 5% flux) for minimization of contact resistance.

### 2.1. Hot Bar Soldering

For the sample preparation a hot bar soldering system from Nippon Avionics Co., Ltd. was used. The thermode used had a square solder surface with an edge length of 1.6 mm. The soldering profile had been developed in a previous test series (Figure 1b). The soldering took 10 s. In this test series, two-sided laser processing of the textile tape was used to strip the wires. The processing on both sides reduced residual insulation in the solder joint and significantly improved the wetting behavior. For every contact point 0.05 g solder paste which contained Sn96.5Ag3Cu0.5 with a metal content of 87.5% and grain size 3 was applied.

### 2.2. Insulation Displacement Connections

For contacting, an insulated conductor was pressed from above into a V-shaped terminal (Figure 2a). This process cuts open the insulation of the conductor and the conductor comes into direct contact with the clamp. The clamp deforms the conductor elastically and plastically by spring forces [20]. The elastic deformation of the conductor leads to the contact force required at the terminal to produce a gas-tight connection. Contact forces can be varied by the material thickness, shape, and the material used for the terminal contact. If the contact force is too low, this can result in an increased contact resistance and the formation of foreign films [21]. American Wire Gauge (AWG) 30 were used for IDCs because IDCs were not available for thinner conductors. We selected IDCs with the designation 1235 from Zierick. The height of the connector was 3.05 mm. The IDCs were conducted with solder (Sn96.5Ag3Cu0.5, 3.5% Flux ROL0) onto the contact pads of the test board (Figure 2b). We used a special press-in tool to connect the IDCs to the textile tape. All contacts could be made in a short time.

### 2.3. Anisotropic Conductive Adhesive

ACAs have significantly lower filler content than ICAs (Figure 3a). The distance between the particles in the insulating polymer matrix is so large that no contact between the particles takes place. This prevents short circuits. At 10–15 μm, the particle size of the fillers is smaller than that of the ICA. The joining of the contact points occurs under pressure, which compresses the adhesive layer to the particle diameter. The specific resistance in the direction of the line is low (10^−4^ Ωcm). Across the line it has a significantly higher resistance (10^14^ Ωcm) which is high enough for many applications [22,23]. The adhesive samples were prepared with an anisotropic conductive acrylate-based adhesive. The contact surface area was measured to be 90 mm², so a force of 135 N was applied. Five UV-light-emitting diodes (395 nm LEDs) were installed into the tool to cure the adhesive (Figure 3b). For curing, a constant light intensity of 400 mW/cm^2^ was used. The amount of adhesive was approximately 0.05 g on each test board. We assumed a shadow by the wires so that the adhesive could not cure completely by irradiation. This is why all adhesive samples were post-cured in an oven at a temperature of 100 °C for 20 min.

### 2.4. Standard Test Methods for Smart Textiles and Their Connections

#### 2.4.1. Continuity Test Before and After Washing

Contact resistance measurement shows the quality of the contact itself. A four-wire measuring instrument, in this case a micro ohmmeter, is used to eliminate connection and line resistances (Figure 4a). The falling voltage at the resistor is measured with a voltmeter over the two remaining conductors. Based on Ohm’s law it is possible to calculate the resistance of the contact. The contact resistances are in the range of only a few milliohm [24]. In order to generate a measurable voltage, high currents of up to 10 A are required.

#### 2.4.2. Peeling Test

A peeling test was carried out to determine the mechanical strength of the joint for further processing. A self-developed peeling test is ideal for inspecting joints between PCBs and conductive textiles. The test boards previously attached for the contact resistance test were clamped into the tensile testing machine in such a way that the conductors peeled off the board at a 90° angle. The force was applied perpendicular to the board (Figure 4b). A load cell with a maximum force of 1000 N was used at a travel speed of 100 mm/min.

## 3. Results and Discussion

Assessment of the contact quality was carried out using several tests. An initial assessment of a successful contact was provided by a visual inspection during a continuity test. A bearable maximum force was determined with a tensile test.

### 3.1. Continuity Test Before and After Washing

Contact resistance provides information about the quality of the contact points. To calculate the contact resistance, a four-wire measuring technique was used by way of the measuring setup from Section 3.4. The wire resistance as well as the contact resistance can be calculated using the formula
R_ges_ = R_1_ + n∙R_L_ ≈ 2 ∙ R_K_ + n∙R_L_(1)
where R_K_ = contact resistance, n = number of wires with the test length of 100 mm, and R_L_: = wire resistance

According to the data sheet, the line resistance for hot bar soldering is 100 mΩ per conductor length (length 100 mm, AWG 32) and 25 mΩ (length 100 mm, AWG 32) for IDC. The line resistance can be determined by measuring different conductor lengths. Due to unreliable contacts the samples with adhesive joints could not be examined in all tests.

Figure 5 shows the nearly identical contact resistances of the soldered and the interlocking contact points. At the same time, information on the line resistances from the data sheets was able to be confirmed by the measurement procedure. After confirming that our measurement method was reliable we needed to simulate a mechanical, thermal, and chemical stress like that which occurs during the use of smart textiles. A resulting load was applied as an example for environmental influences. The samples were washed in a commercial washing machine at 40 °C and retested again. The washing process employed a heavy load because normally the contacts are molded in silicon. If the contact points are unprotected, the loads will be greater than expected during operation. After washing, 75% of the solder samples were faulty. Failure occurred mainly due to the breaking of wires near the soldered joint. This was caused by the solder being drawn into the wire. The drawn solder made the wires crumbly, brittle, and tend to break quickly under mechanical stress. Sixty-seven percent of the specimens with conductive adhesive came loose. This was possibly because of mechanical stress due to the high acceleration forces during the washing process which could not be eliminated due to a faulty bond. Samples with IDCs withstood the washing process. Only one out of 40 contact points failed. The stranded wire broke directly in front of a connector.

Figure 6 illustrates the electrical resistances of hot bar soldering and IDCs after washing. It shows a massive scattering and increase of the contact resistances, especially in case of the soldered joints. In contrast the scattering of the IDCs’ contact resistances can be observed to have increased only slightly. The resistors of the electrical conductors largely remained constant. Observations were:Massive scattering and increase of the contact resistances after washing, especially for the soldered connections.Washing does not influence the wire resistance.

### 3.2. Peeling Test

After examining the contact resistances, the interesting aspect of mechanical strength inspection was considered. We needed sufficient strength of connections for further processing. If the strength was insufficient, the joints could have been damaged before the subsequent spraying process. However, the maximum strength has very little effect on the final strength of the product. The potting compound and the strip structure represent a kind of strain relief for the electrical connection.

### 3.3. Hot Bar Soldering

During the tensile test, three types of failure were observed:Micro cable pulled off at the soldered joint.Micro cables broke off directly at the contact point.Micro cables broke off at the point where the strand draws tin during the soldering process.

In Figure 7, the first force peak indicates the failure of the contacting and the second the removal of the micro cable from the knitted fabric. This occurred when the strands broke up on the side when force was introduced into the textile. On the opposite side they were still connected to the contact points. When the textile was further removed from the circuit board, the strands remained connected to the contact point and were pulled out of the textile. A travel of 50 mm was necessary until all samples were completely cut open. The highest forces and failures of contacting occurred up to a maximum travel of 20 mm, with a comparatively high dispersion of the measured values.

### 3.4. IDC

In most cases the force measurement showed two high points (Figure 7). The connectors were detached in pairs, meaning that with each detachment an increase in force was recognized. When samples were detached from the crimp connection, no strand was cut through. In this case, the failure of one sample did not affect the other units on the textile. Compared to the hot bar soldering test series, the absence of flexible areas in the textile reduced elongation. A movement path of 15 mm was sufficient.

### 3.5. ACA

Specimens with conductive adhesive did not show reliable results in the tensile test. The force increased unstably and constantly to a maximum and then decreased rapidly. Due to the delay in the release of the adhesive, several samples showed force peaks after the first maximum force was applied. Since the adhesive not only bonded the strand at the contact point but also the textile tape with the printed circuit board, it can be assumed that the failure of the contact was already probable before maximum force was reached.

Figure 8 illustrates the maximum forces during peeling tests of test bodies of hot bar soldering, IDCs, and ACAs. In the beginning, it was determined that a force of 5 N was sufficient for further processing of the components. The soldered samples show the highest maximum forces but also the highest dispersion. The form-fit connections failed with the lowest force applied and have minimal scatter. With regard to both parameters, the glued samples lie in between the soldered and pressed samples.

There were clear differences in behavior and bearable strength. Insulation displacement connectors withstood the lowest force average of 8.6 N. The contacts were detached individually. The failure of individual contact points could have occurred before the maximum force was reached. Samples contacted with the conductive adhesive failed on average with a maximum force of 20.53 N. Since the adhesive not only bonded the strand but also the textile to the circuit board, it can be assumed that the contacts failed even before the maximum force was reached. On average, samples from the thermal soldering process withstood the highest load. They failed at an average maximum force of 29.22 N. Since the force curve was continuous up to the maximum force, it can be assumed that the contacting failed in the range of the maximum force. This behavior could be due to elastic elongation and flow at the contact point. All maximum forces determined by the tensile test were found to be satisfactory for further processing.

It can be concluded that the process stability of the IDCs is the highest but that only low strengths were achieved. The large dispersion of the material-locked samples indicates that the process could be further improved.

## 4. Conclusions

The automated contacting of conductors in textiles with an electronic assembly to so-called smart textiles is a technical challenge which needs to be solved by electronic and textile manufacturers. Solution strategies were demonstrated in this work on the basis of a model test for the automated integration of electronic components into a textile. The three most suitable processes from different categories were selected for further investigation, i.e., hot bar soldering, anisotropic conductive adhesive bonding and insulation displacement connection. All of the selected processes are suited for automation. The insulation displacement joint process was the easiest to handle while the hot bar soldering and anisotropic conductive bonding required a significant amount of effort. Process times for contacting ranged from a few seconds for the insulation displacement joint to 30 s per PCB for the adhesive to cure. Hot bar soldering took about 15 s. Insulation displacement connectors have an advantage due to the fixing and the independent positioning of the wires in the connector before the contacting process. At the current stage of development, contacting methods are only conditionally suitable for use in automated production. The bonding process had a weakness due to inadequate contacting. However, it offers high optimization potential with few modifications of the equipment curing. A modified stamp shape and a higher energy density for curing are currently possibilities which could be used to achieve a higher contact rate with a shorter curing time. The contacting of the thermode soldering initially showed greater success. However, it was not possible to prevent the solder from being drawn into the strands by varying the soldering parameters. Drawn-in solder has a negative effect on the textile properties near the contact point. More complex solutions, such as ultrasonic compacting before soldering, have not yet been considered. A greater difficulty than the contacting itself is the correct and automated positioning of the conductors above the contact point. Due to its flexibility and deformability, the outer dimensions of the textile cannot be used to precisely determine the position of the inner conductors. As already described, the strands of the IDCs slip into the contact point itself up to a certain deviation. However, the connectors used are not suitable in practice. No available connectors were found that would allow the original strand size of AWG 32 to be contacted without breaking it up. However, the textile tapes used in the test, which were manufactured with AWG 30 strands, do not exhibit the required textile properties as previously assumed. The development of suitable IDCs is costly but promises the greatest success on the basis of the measurements carried out.

## Figures and Tables

**Figure 1 sensors-20-00005-f001:**
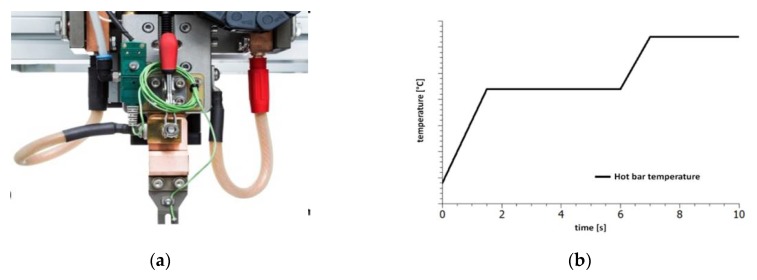
(**a**) Picture of a hot bar soldering system with a 3D thermode and two contact points [17]. (**b**) Temperature profile of a thermode soldering process with a two-phase process; first step: heating and holding to activate the flux, second step: melting the solder and cooling down.

**Figure 2 sensors-20-00005-f002:**
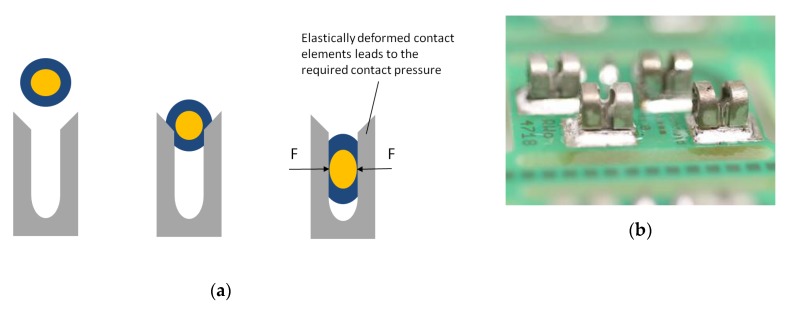
(**a**) Schematic drawing of the assembling of insulation displacement technology: before assembling (left), during assembling (middle), and after assembling (right) in style of Kirstein [4]. (**b**) Prototype of insulation displacement connections (IDCs) placed on circuit boards to integrate the boards in the textile.

**Figure 3 sensors-20-00005-f003:**
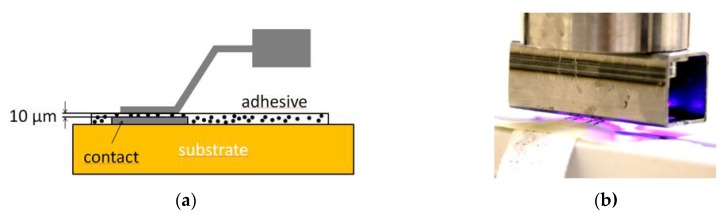
(**a**) Schematic drawing of the contacting mechanism of an anistropic conductive adhesive (ACA); (**b**) stamp for load application and UV curing of the ACA.

**Figure 4 sensors-20-00005-f004:**
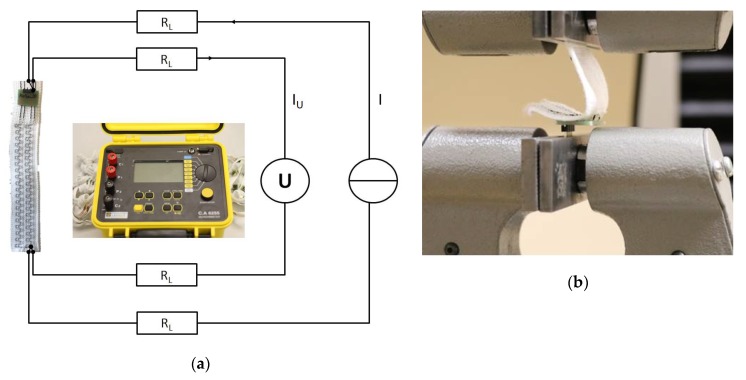
(**a**) Measuring setup for determining the contact resistances using four-wire measuring technology. (**b**) Peel test setup in which the samples were clamped to the blank and pulled on the attached textiles with a speed of 100 mm/min.

**Figure 5 sensors-20-00005-f005:**
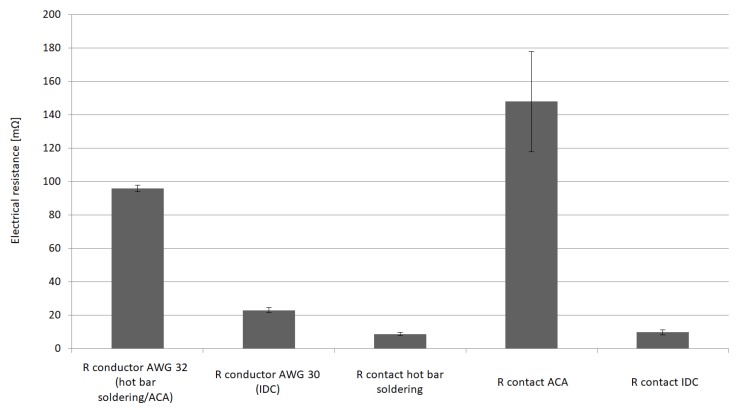
Electrical resistance of hot bar soldered contacts and conductors (AWG 32) compared to IDC contacts and conductors (AWG 30). The contact resistance of hot bar soldering and IDCs is nearly the same. The measured resistance and the resistance from the data sheet correspond, so the measurement method might be reliable.

**Figure 6 sensors-20-00005-f006:**
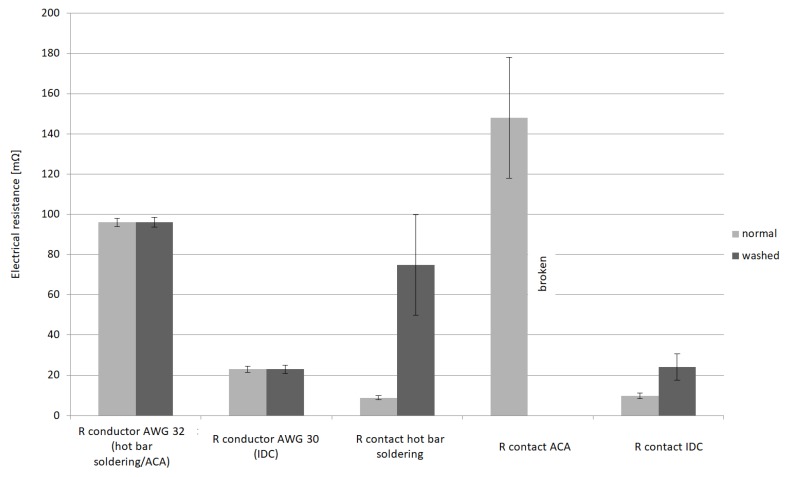
Electrical resistance of hot bar soldered contacts and conductors (AWG 32) compared to IDC contacts and conductors (AWG 30) after washing. The results show a massive scattering of the contact resistances, especially for the soldered connections, while the wire resistance is not really influenced by the washing process.

**Figure 7 sensors-20-00005-f007:**
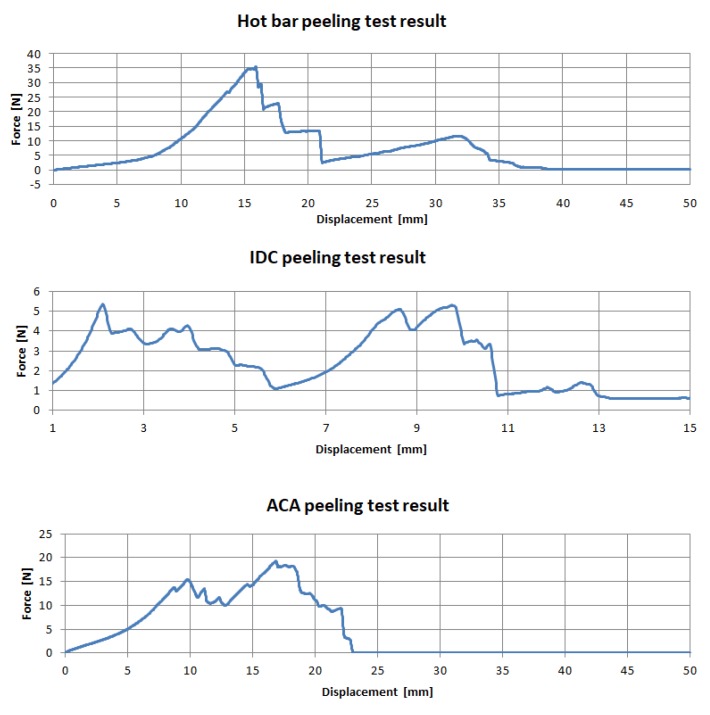
Force-displacement during the peel test. Hot bar soldering (**1**) shows three different failure mechanisms sequentially. In the case of the IDCs (**2**), the contacts failed in pairs. The samples with ACAs (**3**) showed no clear failure mechanism.

**Figure 8 sensors-20-00005-f008:**
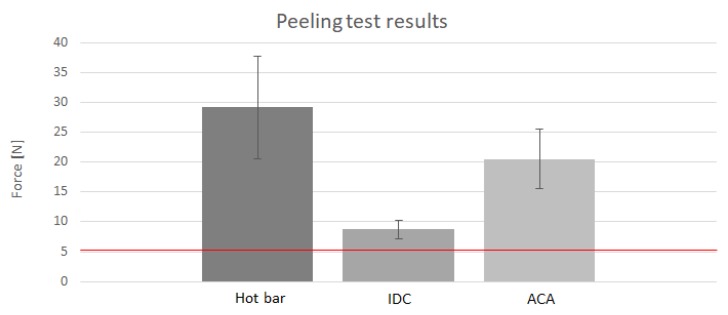
Peace force during peeling tests of hot bar soldered, IDC, and ACA, connections. Hot bar soldering has the highest peak force but also the highest scattering rate.

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
