# Peer review of "Analysis of Hot Bar Soldering, Insulation Displacement Connections (IDC), and Anisotropic Conductive Adhesives (ACA), for the Automated Production of Smart Textiles"

_sensors, 2019, doi:10.3390/s20010005_

Round 1

Reviewer 1 Report

Comments to authors:

Figure 5 and Figure 6, why only the hot bar and IDC methods were performed for comparison rather than included the ACA method? For washing and peeling test, are them the standard methods for evaluating the mechanical strength of the contact resistance of textiles? Please provide the information of the standard test methods. IDC was claimed as the promising method, how to confirm whether the peeling force and mechanical strength are enough for the practical production. It is suggested to simplify the conclusion, some of the contents can be moved into the Discussion.

Author Response

Resubmitting the Revised Manuscript for Sensors

Dear Reviewer,

Thank you very much for reviewing our manuscript “Analysis of Hot Bar Soldering, Insulation Displacement Connections (IDC) and Anisotropic Conductive Adhesives (ACA) for the Automated Production of Smart Textiles” and the opportunity to resubmit a revised copy of the manuscript. We appreciate the interest that you have taken in our manuscript, the complimentary comments and the constructive criticism. We would also like to take this opportunity to express our thanks for your careful and constructive review. Based on the comments, we have made changes of the manuscript, which are detailed below.

We have carefully gone through the comments and tried our best to respond to them. Now we believe this resulted in an improved revised manuscript. In the revised manuscript the changes made are highlighted in red.

We hope that you find our responses satisfactory and that the manuscript is now acceptable for publication.

Best regards,

Sebastian Micus

Reviewer #1 (red):

Comment: Figure 5 and Figure 6, why only the hot bar and IDC methods were performed for comparison rather than included the ACA method?

Answer: ACA method is included

Comment: For washing and peeling test, are they the standard methods for evaluating the mechanical strength of the contact resistance of textiles? Please provide the information of the standard test methods.

Answer: Washing is a very heavy load on the connection between the electronic and the textile and it can happen during the use of Smart Textiles. It is now mentioned in the text.

Comment: IDC was claimed as the promising method, how to confirm whether the peeling force and mechanical strength are enough for the practical production.

Answer: At the beginning, it was determined that a force of 5 N was sufficient for further processing of the components. We added this comment to the results

Comment: It is suggested to simplify the conclusion; some of the contents can be moved into the Discussion.

Answer: The conclusion gives a good overview of the investigated procedures and their advantages and disadvantages. Therefore we would like to keep the conclusion as follows.

Reviewer 2 Report

This paper is quite interesting in smart textiles but there are some areas for improvement before publication. In this paper, there are 3 contacting methods for comparison in terms of continuity test before and after washing and peeling test. There are some uncertainty in relation to changing of tensile properties of textile materials (such as thickness, thermal resistance, softness, etc.) around contact point. In addition, there is no indication of textile materials used in their experiment. Natural fibers and synthetic fibers exhibit different tensile properties. As such contacting method will apply to a particular textile materials, the reader would like to know what the change of tensile properties of textile materials after using contacting method. The result will affect the application of smart textile in the market.

Author Response

Resubmitting the Revised Manuscript for Sensors

Dear Reviewer,

Thank you very much for reviewing our manuscript “Analysis of Hot Bar Soldering, Insulation Displacement Connections (IDC) and Anisotropic Conductive Adhesives (ACA) for the Automated Production of Smart Textiles” and the opportunity to resubmit a revised copy of the manuscript. We appreciate the interest that you have taken in our manuscript, the complimentary comments and the constructive criticism. We would also like to take this opportunity to express our thanks for your careful and constructive review. Based on the comments, we have made changes of the manuscript, which are detailed below.

We have carefully gone through the comments and tried our best to respond to them. Now we believe this resulted in an improved revised manuscript. In the revised manuscript the changes made are highlighted in blue.

We hope that you find our responses satisfactory and that the manuscript is now acceptable for publication.

Best regards,

Sebastian Micus

Reviewer #2 (blue):

This paper is quite interesting in smart textiles but there are some areas for improvement before publication.

Comment: In this paper, there are 3 contacting methods for comparison in terms of continuity test before and after washing and peeling test. There is some uncertainty in relation to changing of tensile properties of textile materials (such as thickness, thermal resistance, softness, etc.) around contact point. In addition, there is no indication of textile materials used in their experiment. Natural fibers and synthetic fibers exhibit different tensile properties.

Answer: The test boards were attached to a knitted tape of polyester fibers (synthetic fibers) with a width of 24 mm and a thickness of 1 mm. It is now mentioned in the text.

Comment: As such contacting method will apply to a particular textile materials, the reader would like to know what the change of tensile properties of textile materials after using contacting method. The result will affect the application of smart textile in the market.

Answer: For contacting, the sensors are attached to the micro cables only. In a further step, the sensors are encapsulated so that the tensile strength of the intermediate product does not say anything about the tensile strength of the end product, which affects the use of smart textiles. In this paper the focus is on different contacting methods for Smart Textiles. For this reason, this value was not recorded.

Round 2

Reviewer 2 Report

Authors have made some improvement on this paper. Although some points have not yet resolved, the author's responses are acceptable. Minor English text editing is required before publication.